# Advances in Electrospun Nerve Guidance Conduits for Engineering Neural Regeneration

**DOI:** 10.3390/pharmaceutics14020219

**Published:** 2022-01-18

**Authors:** Sanaz Behtaj, Jenny A. K. Ekberg, James A. St John

**Affiliations:** 1Clem Jones Centre for Neurobiology and Stem Cell Research, Griffith University, Nathan, QLD 4222, Australia; j.ekberg@griffith.edu.au; 2Menzies Health Institute Queensland, Griffith University, Southport, QLD 4222, Australia; 3Griffith Institute for Drug Discovery, Griffith University, Nathan, QLD 4111, Australia

**Keywords:** fibrous scaffold, neural tissue engineering, structural support, scaffold topography, physical lumen filler, extracellular matrix, peripheral nervous system

## Abstract

Injuries to the peripheral nervous system result in devastating consequences with loss of motor and sensory function and lifelong impairments. Current treatments have largely relied on surgical procedures, including nerve autografts to repair damaged nerves. Despite improvements to the surgical procedures over the years, the clinical success of nerve autografts is limited by fundamental issues, such as low functionality and mismatching between the damaged and donor nerves. While peripheral nerves can regenerate to some extent, the resultant outcomes are often disappointing, particularly for serious injuries, and the ongoing loss of function due to poor nerve regeneration is a serious public health problem worldwide. Thus, a successful therapeutic modality to bring functional recovery is urgently needed. With advances in three-dimensional cell culturing, nerve guidance conduits (NGCs) have emerged as a promising strategy for improving functional outcomes. Therefore, they offer a potential therapeutic alternative to nerve autografts. NGCs are tubular biostructures to bridge nerve injury sites via orienting axonal growth in an organized fashion as well as supplying a supportively appropriate microenvironment. Comprehensive NGC creation requires fundamental considerations of various aspects, including structure design, extracellular matrix components and cell composition. With these considerations, the production of an NGC that mimics the endogenous extracellular matrix structure can enhance neuron–NGC interactions and thereby promote regeneration and restoration of function in the target area. The use of electrospun fibrous substrates has a high potential to replicate the native extracellular matrix structure. With recent advances in electrospinning, it is now possible to generate numerous different biomimetic features within the NGCs. This review explores the use of electrospinning for the regeneration of the nervous system and discusses the main requirements, challenges and advances in developing and applying the electrospun NGC in the clinical practice of nerve injuries.

## 1. Introduction

PNS damage can happen during traffic accidents and other trauma, resections of tumours and/or adverse iatrogenic effects of the surgery occurring in 13–20 of every 100,000 persons [1]. Injury to PNS leads to a severe threat to mobility and sensory function and often leads to permanent loss of function due to the low regenerative capacity of the mature neurons [2]. Despite positive results obtained in the regeneration of damaged tissue in small gap injuries, similar treatments when applied to large gap injuries often yield limited success and result in serious social and economic consequences. To date, the most clinically applied approaches for bridging large gap nerve injuries have relied on surgical procedures, including autografts in which a donor nerve of the patient is resected and grafted into the injury site of the same person [2,3,4]. 

Although autologous nerve grafting is still considered the gold standard, this approach faces fundamental challenges, such as inadequate supply/source of donor cells, loss of function at the donor nerve area, a mismatch between the donor and damaged nerves, and potential painful neuroma formation. Therefore, an effective therapeutic strategy to bring functional recovery is urgently needed worldwide and requires further extensive studies to overcome the current limitations [3,5,6]. As such, studies that focus on the regeneration of dysfunctional neural tissues and restoring or improving lost tissue function have revealed that nerve conduit transplantation might have the potential to be used as alternative therapies to autografts in the treatment of nerve injury [2,7]. 

Nerve guidance conduits (NGCs) are tubular biostructures with engineered biomaterials developed to supply a nourishing and supportive microenvironment for nerve regeneration and to orient axonal growth in a correct path across the nerve injury site [8,9]. NGCs have been used to supply various biochemical and physical factors, including neurotrophic factors, extracellular matrix (ECM) proteins, anisotropic gradients and diverse types of supporting cells in a range of in-vitro and in-vivo models [8]. Due to the ability to engineer NGCs with many different parameters, materials and cells, NGCs offer the ability to be customized to suit each individual injury site and patient variations [6,8]. However, successful implementation of NGCs calls for a fundamental consideration of their structural design, such as a three-dimensional (3D) fibrillar network structure and topographic cues that are capable of up-and downregulation of cell-conduit interactions in the ECM environment [3]. In particular, NGCs with similar morphological, physical, and mechanical properties to ECM might enhance their therapeutic efficacy. Additionally, the greatest potential lies in supplying NGCs with correctly selected biochemical, physical, and biological factors that can work synergistically with other nerve regeneration therapy strategies to activate the growth process [3], including the application of appropriate stimuli such as electrical stimulation [2]. 

This review provides an overview of the requirements for the successful implementation of NGC and the challenges that must be addressed before these therapeutic approaches can be translated into clinical practice for the treatment of PNS injuries. The review suggests potential solutions to overcome the current limitations, which will also allow the development of the next generation of therapies.

## 2. Point to Consider for Developing NGCs 

### 2.1. The Conditions of the Injury Site

The primary considerations in the treatment of nerve injuries are understanding the pathomechanisms behind nerve damage and associated soft tissue or vascular injuries [6]. The cellular and molecular responses to nerve injuries are dependent on many factors, particularly the location of the nerve, the time elapsed, and patient age [7]. With PNS injuries, physical trauma can lead to the death of the axons, demyelination of the supporting Schwann cells and production of myelin debris, and the infiltration of macrophages in response to changes in chemokine and cytokine secretion. Transplantation of NGCs into this hostile injury site must therefore consider that considerable cellular debris exists, and that the molecular environment may not be conducive to supporting growth of transplanted cells [10,11]. However, due to the numerous bioengineering characteristics that can be used, it is possible to design NGCs that can modulate various aspects of the injury site and thereby enhance integration and regeneration.

### 2.2. General NGC Requirements

From a clinical research standpoint, there are numerous physical characteristics that need to be considered. The NGC needs to be flexible so that it can be easily handled by surgeons during implantation without excessive manipulation or damage to the surrounding tissue. After implantation, the NGC needs to be non-toxic/biocompatible with the cellular environment and not stimulate any mutagenic, carcinogenic or cytotoxic behaviour, and be minimally immunogenic so that it does not exacerbate inflammatory responses [12]. The NGC needs to be permeable to some degree to facilitate fluid exchange but should have minimal swelling. Ultimately, the NGC should be biodegradable so that it does not require surgical removal. However, regulating the degradation rate is critical as the outcome of conduit transplantation can be negatively affected by too slow or too rapid a degradation process [13]. If the degradation occurs too quickly, then the regenerating axons may fail to cross the injury site, and if the degradation takes too long, then the remaining components of the NGC may hinder the endogenous cell reorganization and regeneration. In order to obtain the optimized degradation rate, various chemical and physical properties, including molecular weight, chemical structure, degree of crystallinity of the applied biomaterials, have to be considered. Additionally, the degradation rate could be influenced by the morphological structure features of the NGCs such as size, shape, density, and porosity, to name but a few [6,9,13,14]. As such, the optimum balance of the aforementioned features should be taken into consideration for the successful application of NGC in nerve regeneration therapy. 

The mechanical properties of NGC also play a key role in the repair of nerve gaps. Some of the mechanical roles of an NGC is to create a barrier to protect new axons from the encroaching scar tissue, to prevent the surrounding tissue compressing against the regenerating cells and to provide stable structural support until there is sufficient regeneration of the nerve. In addition, the NGC also has to resist tearing from sutures if they are used. As peripheral nerves are subject to stretching, compression, and shearing forces, the NGC must also be flexible and have the ability to withstand these forces [12,15]. To generate NGCs of appropriate strength and flexibility, the mechanical properties of the target area have to be estimated [5] and, therefore, special attention must then be given to the selection of materials so that they match the required mechanical characteristics of the target nerves. 

Size mismatching is a major challenge associated with the use of autografts [16]. The use of NGC has been investigated as a solution since they can be fabricated into size-matched structures. A successful NGC design have to specific to recipients and requires a concise consideration of the location of the nerve repair and size of the nerve gap. NGC morphological structure is another important area affecting nerve regeneration. For example, a lower wall thickness increases neuroma formation [4]. Additionally, the optimal internal diameter allows the damaged nerve to grow without compression while inhibiting ingress of surrounding tissues [8]. Therefore, optimizing NGC design before translating requires further extensive investigations. In this regard, rapid prototyping methods are leading towards advances in customized and personalized NGC design [16]. All the aforementioned factors are basic requirements in creating NGCs with the ability to provide the appropriate micro-environment to withstand the numerous molecular, cellular, and physical challenges of nerve repair [6].

### 2.3. NGC Structure

The morphological structure of NGCs plays an essential role in the successful regeneration of the injured nerve. As axons need to extend through the NGC, it is important that the morphological structure has a well-aligned orientation but also has porosity to offer axons the ability to seek new paths that may be more appropriate. Porosity is also important for allowing nutrient exchange and determining the appropriate balance of porosity versus structural alignment is one of the challenges [3,9]; as with too much porosity, axons may wander too widely and form a neuroma, whereas, with too much alignment, axons may be forced into inappropriate directions and therefore fail to make functional connections. Perhaps the most useful morphology to replicate is that of ECM, which is composed of a nanoscale network of proteins and glycosaminoglycans. These networks can make a barrier between tissues and provide a supportive meshwork around the cells to supply cell anchorage [10]. Therefore, studies have tried to develop scaffolds with similar features to ECM at the nanoscale level.

The manufacturing method is critical, as it can influence the resultant structure. There have been considerable advances in the range of methods that can be used to manufacture NGCs, including freeze-drying or lyophilization (low-temperature dehydration process under a vacuum), self-assembly (a process of association of system’s pre-existing components into an ordered structure or pattern), solvent casting (a process of manufacturing by immersing a mould in polymer solution), gas foaming (a high-temperature process for the production of foam-based polymer scaffold), and 3D printing (a creation of three-dimensional compartment under computer control) [14]. Despite the advantages associated with some of these techniques, the majority of them are incapable of developing nanofibrous substrates that mimic ECM, which is composed of fibres ranging in diameter from 50 to 500 nm [12]. For example, self-assembly is not scalable, and control over fibre dimensions is challenging. The low interconnectivity of porosity, small pore size, and irregular porosity are the main drawbacks of both gas foaming and freeze drying. The low mechanical strength limits the 3D printed scaffolds [10]. Being able to produce scaffolds that have various nanoscale fibres that mimic ECM may have the potential to increase cell-to-scaffold interactions and increase nutrient exchange that can stimulate the regeneration of neurons and associated supporting cells [17]. 

Among the aforementioned fabrication methods, electrospinning has generated considerable advances due to its ability to produce fibres of various scales and which have a large surface area with a three-dimensional (3D) porous structure resembling the native ECM network [10]. As such, electrospinning is one of the most widely studied techniques in fabricating nanofiber conduits [18,19]. Electrospinning offers the advantages of being a simple, low cost, controllable, and well-established technique for fabricating various fibrous meshes in different forms. Due to its ability to regulate the directional flow, it can create fibres of different orientations such as random fibres, aligned fibres, 3D fibrous scaffold, and core-shell fibres [20]. It also can use a wide variety of materials to produce continuous fibres with a range of properties, particularly those that confer the desired mechanical properties [12]. 

The electrospinning process can generate the various electrospun fibres via controlling the properties of the solution, processing, and environmental factors such as humidity and temperature (See Figure 1) [10]. Numerous parameters can affect the production process, including the polymer molecular weight, concentration, viscosity, electrical conductivity, elasticity, polarity, and surface tension in polymer solutions. From a fabrication point of view, the feed flow rate, applied field of voltage, nozzle-to-collector distance, the geometry of round collector, and collector rotation speed have key roles in generating the structure and alignment of the nanofibrous biomaterials [10,21,22]. As such, the diameter and the morphology of electrospun nanofibers are affected by different properties of electrospinning solution, process, and environment [10]. For example, higher-molecular-weight polymers and polymer concentrations in electrospinning solute result in larger fibre diameters, owing to higher viscosity and surface tension in polymer solutions during electrospinning, whereas the increase of applied voltage at the nozzle and the temperature results in decreasing fibre diameter [10,21].

In NGC fabrication, the repeatability and reproducibility of the synthesizing process is key. Although electrospinning has been demonstrated as a well-suited technique capable of producing desired nanofibers via modifying electrospinning process parameters, obtaining identical properties, such as pore size, porosity, and fibre direction, are still major challenges and require precise controlling over the fabrication process and ambient parameters [10,16,23]. 

### 2.4. Application of Electrospun Substrates for Repairing the Nervous System

Numerous nanofibrous scaffolds have been extensively used in neural regeneration applications. For example, Debski et al. examined a nanofiber-based nerve conduit composed of poly (L-lactic acid)-co-poly(caprolactone), collagen and polyaniline (PANI) in a rat model. This conduit presented muscle atrophy decrease and was suggested as a mean for axonal regeneration support and managing nerve gap as in peripheral nerve [24]. Polycaprolactone/collagen VI electrospun conduits were assessed by Lv et al. in a 15-mm-long sciatic nerve defect in rats and reported the sustained release of collagen VI enhanced the recruitment of macrophages and their polarization toward the pro-healing (M2) phenotype [25]. Another study of the application of nanofibrous in nerve conduits offered polyvinyl alcohol (PVA)/carbon nanotubes (CNT) electrospun films as a suitable material for nerve conduits [26]. In a recent work by Wu et al., two different fabrication methods, lyophilization and electrospinning, for preparing chitosan scaffolds were compared for Schwann-cell transplantation in rat models, and they reported the electrospun scaffolds more favourable for cell–cell interactions in PNS repair [27]. Table 1 lists recent studies to examine the application of electrospun nerve conduits for repairing the nervous system.

As previously mentioned, for successful application of electrospun scaffolds in NGC, these fibres should meet some morphological requirements, including appropriate diameter sizes and porosity distribution. The fibrous meshes obtained by electrospinning show a range of 100–1100 nm fibre diameter. Fibre diameter is known to have an impact on neural cell growth and Schwann-cell (SC) migration. To illustrate, a work by Wang et al. reported the neurite length on scaffolds with the intermediate and large fibres were higher than those with thinner fibres [28]. However, the optimum size has to be measured; for example, another study reported that Schwann cells presented lower elongation lengths on the fibrous scaffolds with average diameters of 5 and 8 µm compared with those with an average diameter of 1 µm [29]. It also reported that scaffolds with larger fibre diameters could allow increased cell penetration [10]. 

**Table 1 pharmaceutics-14-00219-t001:** Recent studies focusing on the use of electrospun nerve conduits for regenerating peripheral nerves.

Biomaterial	Cells	ES Parameters			Stimulating Agents	Stimulating Patterns	In Vivo	Refs.
		Voltage (kV)	Flow Rate (mL·h^−1^)	Distance (cm)				
chitosan	Schwann cells	4	3	-	BDNF & VEGF	aligned fibres	sciatic nerve defects in rats	[30]
PLA/PPy	rat hippocampal progenitor	15	-	10	PPy-coating	external stimulus (200 mV/cm)	-	[31]
PCL/chitosan	Schwann cells, PC12 cells and dorsal root ganglia	15	1.5	-	-	aligned fibres	sciatic nerve in adult female Sprague–Dawley rats	[32]
PLCL	murine macrophage cell line and rat Schwann cells	16	2	10	-	oriented microfiber-bundle cores and randomly organized nanofiber in wall of NGC	rat sciatic nerve injury	[33]
PCL	Schwann cells	14	0.2	-	sodium alginate hydrogel covalently cross-linked with N,N′-disuccinimidyl carbonate (DSC)	bilayer cylindrical conduit	sciatic nerves in a rat model	[34]
polyvinyl alcohol (PVA)/carbon nanotubes (CNT)	fibroblasts	19 and 21	0.06–0.08	10	-	providing conductivity via CNT	-	[26]
poly (lactide-co-trimethylene carbonate) (PLATMC)	Schwann cells	-	-	-	-	shape memory nanofibers	rat sciatic nerve defects	[35]
poly (L/D-lactic acid) (PLDLA) and phosphate glass microfibers (PGFs)	dorsal root ganglion	1.5 kV cm^−1^	0.1 mL min^–1^	-	CNTs chemically attached on the surface of the NGC	-	transected rat sciatic nerve	[36]
PCL	bone marrow stem cells (BMSCs)	12	1	-	-	honeycomb structure	-	[37]
PLLA	dorsal root ganglion	15	1	10	porcine decellularized nerve matrix hydrogel	aligned fibres	rat sciatic nerve defect model	[38]
PCL	PC-12	11	0.25	5.5	cross-linking laminin	aligned fibres	rat sciatic nerve gap	[39]
poly (lactic-co-glycolic acid) (PLGA)	-	10	0.4	15	collagen sponge	intraluminal sponge fillers	rat sciatic nerve	[40]
poly (L-lactic acid)-co-poly(€-caprolactone), collagen (COL), polyaniline (PANI)	adipose-derived stem cells (ASCs)	15	1	8	-	-	rat model	[24]
PCL/collagen VI	macrophages	15	4	18	sustained release of collagen VI	-	rat sciatic nerve	[26]
chitosan	Schwann cell	15	1	10	-	-	-	[27]
PVA/gelatin/gellan	neural cells	19	0.8	15	quercetin	patterned hybrid of aligned fibres scaffold	-	[41]

Creating proper porosity size and distribution to allow entry of nutrients and waste removal and guide axonal growth is another important area in NGC design. It is reported that the NGC with high porosity (>80%) supplies enough permeability for the nutrient flows from the outside to the inside of the conduit leading to improved nerve regeneration and simultaneously preventing infiltration of unwanted tissue into the NGC [4]. However, worth to mentioning much uncertainty still exists about selecting a proper method that is able to produce precise measurements of the porosity through an electrospun mesh.

### 2.5. NGC Material

In addition to the nanofibrous structure of NGC, their functionality and effectiveness are strongly influenced by material selection. The definition of the American National Institute of Health (NIH) for biomaterial is “any substance or combination of substances, other than drugs, synthetic or natural in origin, which can be used for any period of time, which augments or replaces partially or totally any tissue, organ or function of the body, in order to maintain or improve the quality of life of the individual” [42].

The extensive research in tissue engineering has been indicated that correctly chosen biomaterials can support the integrity and regeneration of cells without inducing inflammation [1]. Materials of the NGC should support the physical, chemical, and biological environment surrounding the neural and glial cells [7]. Both natural and synthetic biopolymers have been used in NGC production. Natural polymers possess a diverse property reflective of naturally occurring tissues and are popular carriers in tissue engineering. For instance, collagen is an abundant structural protein of various connective tissues, and its desirable properties, such as high biocompatibility, make it a popular natural polymer for NGCs [20]. Chitosan, silk fibroin, and extracellular matrix components are other natural materials utilized for neural scaffolds [43]. However, there are several limitations with the application of natural biopolymers, including the variability of mechanical properties, low environmental stability, cell-mediated immune responses, and risk of infection. In contrast to natural biomaterials, synthetic ones can be designed so that they have higher mechanical strength and stiffness, and the fabrication process can be tightly controlled to ensure uniformity [44]. Silicone was one of the earliest synthetic biomaterials used for synthetic nerve conduits mostly due to its elasticity. However, poor nutrient transfer, fibrotic host response and the irritation at the site of surgery, as well as the need for scarred tissue removal, were the common drawbacks of the silicon-based devices [20,45]. 

A wide range of synthetic polymeric materials has been used in nerve conduit fabrication among which conduits constructed from aliphatic polyester-based polymers, such as polycaprolactone polylactic-co-glycolic acid (PLGA), polyglycolic acid (PGA), poly(ε-caprolactone) (PCL) and poly (DL-lactic acid –co-ε-caprolactone) (P(DLLAco-CL)) have been investigated thoroughly as promising candidates in some clinical trials [5]. However, for many synthetic biomaterials, a considerable drawback is that cells do not readily attach to the synthetic biomaterial, and inflammatory reactions to the biomaterial material can limit their use. In addition, biomaterials with high rigidity can cause mechanical trauma at the injury site, fistula formation, and extrusion, which drive the search for variations in structural properties to identify synthetic biomaterials with more flexibility. Considering that natural and synthetic biomaterials offer benefits in different ways, a solution can be the combination of natural and synthetic biomaterials to work synergistically to provide an optimal mix of biocompatibilities, desired mechanical properties and degradation patterns [20]. For example, the natural polymer in a composites scaffold can improve the biocompatibility and biodegradability of the mixture via providing a biochemical interaction between cells and the scaffold [46]. An alternative strategy is to add some natural polymers to the surface of synthetic scaffolds, which can aid cell attachment, since the upper-most surface of the scaffold can have a major impact on cell attachment [44].

### 2.6. NGC Surface

A surface of the scaffold that promotes the adhesion of cells is key in achieving functional cell/scaffold interactions. Numerous characteristics can impact the biochemical mechanisms and the properties of the surface to influence cell adhesion, including the topographical features, stiffness, functional groups, hydrophilic/hydrophobic properties, and interfacial free energy [10,44]. As such, changing the surface chemistry, including bioactive molecule immobilization, has been broadly applied on the biomaterial surface. For example, gelatine treatment of an electrospun poly (lactic-co-glycolic acid) (PLGA) conduit improved the adhesion of mouse embryonic stem cells [14]. Additionally, surface modification with polydopamine has been shown to improve the hydrophilicity and stability of the material surface, leading to effective cell growth, adhesion, proliferation, and differentiation relevant for applications for peripheral nerve regeneration [47]. Functionalizing with conductive compounds, crosslinking of nano bioglass, and nerve growth factor immobilization are other common chemical surface modification approaches [48].

Apart from providing chemical cues, cellular behaviours such as adhesion, migration and differentiation, as well as the regeneration of new tissues, can be influenced by topographical characteristics [49,50]. Cells’ receptor clustering or curvatures on the cell membrane can be affected by properties of the scaffold; therefore, imposed surface topography can have a huge impact on cellular responses, organization, and function, and have been reviewed in detail in the following reviews [7,51,52]. For example, the surface stiffness (or hardness) and the surface roughness have ability to affect the secretion of specific channel proteins from the cells leading to induce the desired signalling events and control neural cell development [3]. Thus, the topographical modification of the scaffold, such as changing the types, sizes, and spacing of surface patterns can alter surface energy improving the adsorption and bioactivation of the ECM proteins. In-depth consideration of the topographical modification effects on the nerve regeneration process is beyond the scope of this study. In this regard, the effect of topological structures in peripheral nerve repair was extensively reviewed by Ma et al. [53]. It can be concluded that providing proper contact guidance via chemical and topographical cues can be an important area of NGC design.

### 2.7. NGC Topographic Structures

To date, NGCs have been fabricated in a range of structures, such as cylindrical tubes with internal channels or intraluminal guidance, porous walls with electrospun outer conduits, or combinatorial techniques [14]. Earlier works fabricated NGCs with a single hollow shape to resemble the tube-like structure of nerves, which mainly fabricated by injection moulding, melt extrusion (a melted polymer is extruded within the nozzle), physical film rolling (a polymer mat is rolled around a mandrel, the edges of the roll are overlapped, sealed and compressed), braiding, and crosslinking (a cross-linking agent adds to a polymer mixture then loaded into a cylindrical mould) [9,16,43]. However, without internal architecture, regenerating axons were unable to navigate appropriately and often, the distribution of axons across the graft was limited with the result that the axons became misdirected and did not innervate their appropriate targets or the axons branched and innervated multiple targets [3,54]. These limitations led to the design of NGCs with an architecture that mimics the natural structure of nerve organization [15,43]. In this regard, multichannel nerve conduits have typically achieved better outcomes in axonal regeneration, compared to nerve conduits with a single lumen; as, with a higher internal surface area available, there is increased cell adhesion and migration and reduced axon dispersion [55]. A study focusing on developing NGC with a similar structure to the PNS anatomy designed a multi-tubular conduit made of electrospun polycaprolactone (PCL) fibres with a honeycomb structure seeded by BMSCs. They reported the BMSCs migrated and proliferated in all the small tubes and transdifferentiated into Schwann-like cells [37]. 

However, due to the multilayers or channels within these complex conduits, there are additional limitations that need to be overcome. Fluid permeability and nutrient exchange can be low, particularly in the internal regions, and the structural rigidity can be higher, which can increase cell death rates due to the lack of adequate nourishing sources and the cell metabolic waste removal [3,43]. 

To overcome some of the challenges faced in the implantation of the multichannel conduits, NGCs have been produced with physical lumen fillers: physical fillers, which fill the internal space of the nerve tube. A wide range of filler materials with a diverse geometrical property, such as hydrogel matrix (typically made of polysaccharides, ECM molecules, proteins, and peptides) and micro-/nanofilaments (fabricated via electrospinning and rapid prototyping) have been used as physical lumen fillers (See Figure 2) [56]. Among these fillers, microfilaments and nanofibers offer the possibility for resembling the nature structure of the nerve and have the potential to be used as filler material in the NGCs [57]. This generation of NGC represents a promising frontier in nerve repair, where neurites grow and orient with the incorporated luminal fillers. As the regenerating axons and supporting, cells can grow across and along all regions of the lumen filler, there is more opportunity for the axons to sort out and navigate according to axon guidance cues [43]. 

## 3. Tissue-Stimulating Agents and/or Patterns

A fundamental consideration in creating a neural scaffold is its ability to provide a cell anchoring site that can then stimulate tissue formation and cell function. To achieve this, the structural design of the carrier and the physiological niche need to complement each other so that the combined effect creates a supportive microenvironment [10,58].

To date, diverse regenerative cues have been used for neural tissue engineering to work synergistically with other NGC features such as material and morphological structures [11]. Below, a brief summary of the most common uses of these potential therapeutic molecules employed in nerve repair is provided.

### 3.1. Incorporated Cells

While nerve conduits alone may stimulate endogenous cells to regenerate the injured nerve, results to date suggest that incorporating cells into the nerve conduits can offer additional therapeutic benefits [59]. After a major injury, two different cell transplantation strategies can be used. To preserve or replace lost neurons, cell transplantation approaches can provide both neuroprotection to cells before apoptosis as well as cell replacement to restore functionality at the implantation site. While damaged neurons can be replaced by transplantation of neural stem cells, the cell bodies of peripheral nerves are often contained within discrete ganglia, and, thus, there can be limited opportunity for the successful replacement of lost neurons. Alternatively, transplantation can replace the supporting cells of peripheral nerves, the glial cells (predominantly Schwann cells) and perineural fibroblast either by using differentiated cells or progenitor cells that can be driven to become the mature target cells. By repairing the supporting cellular structure, the endogenous axons can then navigate through the injury site. However, functional recovery in the target area is closely dependent on proper interactions among cells, which are recruited to the injury site to aid clean up and repair [25]. Thus, the treatment of PNS injuries can involve the transplantation of several different cell types [59]. However, these cell-based therapeutic approaches need a robust source of healthy cells, and while Schwann cells and fibroblasts can be harvested from donor nerves, a potential alternative is the use of stem cells with the potential to produce neural cells, including neurons and glia [44,60]. Mesenchymal stem cells (MSCs) are derived from various tissue sources and have self-renewal potential and the ability to differentiate into diversified cell types. MSCs have been broadly investigated in a range of in-vitro and in-vivo models, with reports of positive regeneration outcomes after peripheral nerve injury [58,60]. In this context, bone marrow stem cells (BMSCs) have been demonstrated to modulate Schwann cell behaviour resulting in enhanced repair [61]. It is not just the cells themselves that can exert positive outcomes, but also cellular components. Exosomes from BMSCs have been incorporated with a NGC composed of the aligned electrospun polyurethane fibres in the repair of the PNS injuries after diabetic peripheral neuropathy [62]. 

However, restoration of full nerve function requires the repair of all the cellular components of a peripheral nerve. For this reason, special attention is given to the transplantation of Schwann cells (SCs), which are the glial cells of most peripheral nerves. SCs are the major glial cells of the PNS and have many roles, including maintenance of the nerve, myelination of the axons, and secretion of numerous molecules that promote axon growth and nerve regeneration, including neural cell adhesion molecules (N-CAM), collagen, laminin, and adhesion molecule L1 [3,60,63]. SCs at injury sites secrete trophic factors that control the regeneration process of axotomized neurons [64]. When transplanted into a peripheral nerve injury site, the SCs can migrate across the injury site to form a cellular bridge across which the regenerating axons can be guided (See Table 1 for some studies evaluate incorporation SCs with electrospun NGCs). 

An alternative glial cell type is olfactory-ensheathing cells (OECs), which have similar properties to SCs, but also offer additional characteristics and benefits for neural repair. Similar to SCs, they can enhance the microenvironment after nerve injury via secretion of neurotrophic factors including brain derived neurotrophic factor, glial derived neurotrophic factor, and nerve growth factor [65,66], as well as extracellular matrix molecules, which together can indirectly activate endogenous SCs [60,67,68,69]. Compared to SCs, OECs express higher levels of macrophage migration inhibitory factor (MIF), which prevents the infiltration of macrophages into the injury site and thus OECs can modulate the inflammatory response [65,66,70]. 

However, for both SCs and OECs, the transplanted cells need to be contained within a scaffold so that they can be easily handled by surgeons during implantation and so that the migration of the glial cells can be directed along with the nerve injury site. Therefore, NGCs need to be identified that are not only compatible with MSCs, SCs, and OECs but which can promote the robust functional activities of the incorporated cells.

### 3.2. Neurotrophic Factors

As mentioned above, glial cells secrete a range of factors that support cell survival and growth, promote axonal regeneration and functional recovery following the nerve damage [45,61]. These secreted factors can themselves be incorporated into NGCs and thereby exert additional regenerative benefits. A range of functionalized electrospun nerve scaffolds has been used in models of in-vitro and in-vivo nerve injury [4]. During NGC fabrication, neurotrophic factors can be directly conjugated to conduit walls and/or located in the lumen [8]. However, the effectiveness of the NTF is limited by several different parameters, including their relatively low stability and poor release kinetics, and a lack of understanding of the appropriate dosing that should be used at the target area [8].

### 3.3. Extracellular Matrix Proteins

The combination of NGC with extracellular matrix proteins, such as collagen, laminin, hyaluronic acid and fibronectin, has been used to replicate the natural ECM and to interact with cells to drive improved regeneration [45]. As such, it was reported a combination of these ECM molecules as a filler of the NGC can enhance the regenerating environment and consequently increase effective regeneration [71]. In addition to providing an internal surface for neural regeneration, the ECM proteins can also be coated on the external surface of the implants to increase adhesion and integration into the host tissue [45,71].

### 3.4. Electrical Conductivity and Stimulation

Emerging studies suggest that conductive biomaterials are appropriate candidates for fabricating functional nerve conduits [2]. The nervous system functionality relies on the transmission of electrical signals. During regeneration of peripheral nerves, the re-establishment of electrical connections can drive further regeneration of the neurons as the positive feedback of successful targeting leads to further trophic support. Facilitating the electrical signalling by generating carriers with electrical properties can be effective in driving neural action potential and consequent nerve regeneration [15]. Some of the common conductive biomaterials used in tissue engineering applications include graphene, carbon nanotubes (CNT), polyphosphazenes, polyaniline (PANI), polypyrrole (PPY), and polythiophene (PT) [2,54]. Similar to the various additive components mentioned above, there can be dual applications within the NGC; in addition to the benefits that would be provided by electrical conductivity within the lumen or scaffold, coating the exterior surface of fibrous scaffolds with conductive material has been shown to lead to enhanced hydrophilicity of the surface and higher cell attachment [14]. 

The efficacy of the conductive NGC can be improved in the presence of electrical stimulation. External electrical stimulation can also improve neurite outgrowth [10]. The electrical stimulation offers the possibility for enhancing muscle preservation and functional outcome. A recent study investigated the electrical stimulation effects of co-woven NGCs made of electrospun poly-(L-lactic acid) (PLLA) yarns and PPy-coated PLLA yarns and reported the enhanced neurite outgrowth of rat dorsal root ganglion (DRG) neurons on this NGC [72]. Research studies have shown that organic piezoelectric polymer such as PVDF (polyvinylidene fluoride) may be a promising candidate for NGC due to its strong piezoelectric electrophysiological properties in the presence of strong electric field polarization leading to enhance the cell activity and function. Electrospun PVDF scaffolds also suggested as a powerful mean with the ability to direct neural stem cells differentiation into neuronal and glial cells [73]. Another study of the application of Electrospun PVDF in PNS regeneration offered electrospun nanofibers composed of piezoelectric polyvinylidene flouride-triflouroethylene (PVDF-TrFE) that promote the adhesion and alignment of Schwann cells and fibroblasts [74].

However, some conductive biopolymers, such as polypyrrole (Ppy), are non-biodegradable and requires a second follow-up surgery to remove the NGC [5]. In recent years, bioresorbable electronic stimulators have been introduced as a novel technology with the ability of long-term electrical stimulation leading to improved functional recovery. These bioresorbable electrical stimulation platforms overcome the issues associated with traditional implantable stimulating devices such as discomfort, pain, costs, and the need for the second injury to retrieve them [75].

### 3.5. Magnetic Properties

Magnetic fields can influence axon guidance, and the incorporation of magnetic nanoparticles, such as iron oxide, within neural scaffolds that are then subjected to an external magnetic field can increase the expression of BDNF, GDNF, NT-3, and VEGF by SCs [76]. Magnetically responsive nanoparticles in the presence of a magnetic field have been shown to regulate SC biological activities [77] and, as such, magnetic fields have emerged as a feasible modality in stimulating axonal growth and promoting PNS regeneration [76,77,78]. For example, Lacko et al. demonstrated the incorporation of iron oxide nanoparticles with hydrogel scaffolds could guide cellular migration and axon regeneration [79]. As such, the orienting and stimulation of axonal growth through tension forces by external magnetic fields is suggested as a key area of focus for future research. In this context, synergic effects of physical stimuli, such as nano surfaces and magnetic nanoparticles, can facilitate nerve regeneration. For a more detailed review of these concepts, see [80].

### 3.6. Alignment of Fibres

The extensive research history behind the structural requirements of electrospun fibres indicated that the alignment of the scaffolds can influence cell proliferation and growth and guide axonal growth [14,21,81,82]. Figure 3a,b show SEM images of aligned fibrous scaffolds at two different resolutions. The quantitative distribution of nanofibre diameter and orientation direction are presented in Figure 3c,d, respectively. Figure 3e,f shows the SEM images of the neurons on the aligned scaffolds. These two figures show that neurites extend and align themselves along the scaffold fibres, which was also quantitatively assessed by ImageJ^®^ analytical software (National Institutes of Health, Bethesda, MA, USA) and shown in Figure 3g. A dominant peak at around 90° in this directional histogram of cells on the aligned scaffolds confirms the neurite guidance capacity of these structures. As such, electrospun conduits with an aligned structure have been shown to be superior to the hollow tube conduits, owing to their high flexibility, porosity and high specific surface area resulting in increased protein absorption, Schwann-cell migration, and axon generation [14]. Scaffolds with aligned fibres have been tested and in one example, an aligned chitosan nanofiber hydrogel grafted with peptides aided repair of sciatic nerve defects in rats [30]. In another study, aligned polyhydroxyalkanoate electrospun scaffolds are reported as a potential inner structures of nerve conduits by providing neurons with efficient growth and differentiation [83]. Importantly, the alignments of fibres can be more effective on the uppermost layer, and cells may not be able to penetrate into the deeper layers of the scaffold. This was observed in tests of a double-layered electrospun NGC with aligned and random nanofibers on the top and bottom layers, with the morphological structure of Schwann cells only influenced by the topmost layer of fibres [84]. 

### 3.7. Combination of Stimulating Strategies

A synergistic effect of combining aligned electrospun fibres with conductive additives has been a focus of recent investigations as a prominent means of bio-mimicking the native nerve with the aim of facilitating nerve regeneration and functional recovery. 

Topological guidance such as alignment can also work synergistically with biochemical factors in nerve reconstruction. An example of this combination is where NGCs made of aligned PLLA nanofibers were incorporated with porcine decellularized nerve matrix hydrogel (pDNM gel). When used to repair rat sciatic nerve defect models, the combination NGC resulted in an increased number of axons, extensive myelination and improved sciatic nerve function in both in-vitro and in-vivo tests [38]. In a recent investigation into developing an NGC that can control neural cell directional growth, aligned nanofibers made of PVA, gelatine and gellan were further processed to make a patterned hybrid scaffold which was then incorporated with quercetin to obtain functional contact guidance for neural cells. Quercetin offers antioxidant and free radical scavenging properties and can prevent oxidative stress-related diseases, such as neurodegenerative disorders. This study reported that the proposed hybrid NGC enhance the regeneration process compared to the scaffold made of aligned electrospun nanofibers alone [41]. 

## 4. NGC in Clinical Trials

The clinical application of NGCs dates back to the nineteenth century. Different hollow tubes, such as decalcified bone and vessels from human and animal origin, have been applied to fill the gap caused by nerve damage [85].

In recent years, conduits have been available in various configurations and diameters and prepared from natural and/or synthetic polymers to use in the clinic [86]. Numerous commercial guidance conduits and wraps for PNI are now available for preclinical and clinical research, including Avance Nerve Graft, AxoGuard Nerve Connector, AxoGuard Nerve Protector (produced from ECM derived from porcine intestinal submucosa), Reaxon (made of chitosan), NeuroMend (made of type I collagen), NeuroMatrix (made of collagen), NeuroFlex, NeuraGen (made of collagen), NeuraWrap, Neurolac (made of PLCL), NeuroTube (made of polyglycolic acid (PGA)), and SaluTunnel (polyvinyl alcohol (PVA)) [20,45,86]. Some of these use combination approaches such as Nerbridge (Toyobo, Osaka, Japan), which is a commercial PGA-based NGC with microtube guidance channels that are filled with an inner collagen matrix [45].

In spite of the promising results from the most recently approved devices, effective repair of peripheral nerves is still inadequate and it is clear that more detailed characterization of the functionalities and comprehensive comparative assessments with the standard gold autograft is necessary to develop these NGCs as an alternative [86]. In a recent study, nerve conduits and nerve graft were compared and found patients with 11–17.99-mm lesions presented significantly bigger improvement in grafts compared with NGCs. However, this work claims that identification of the better treatment is still impossible due to insufficient data [87]. Another work by Saeki et al. examined the efficacy and safety of collagen NGCs filled with collagen filaments and reported that a rate of discovery were 75% for the NGC group and 73.7% for the autologous group in nerve defects of 30 mm [88]. There is a range of clinical trials currently underway or finalized the efficacy of the different NGCs. For a more detailed review of these concepts, see [6,87,89,90].

## 5. Future Trends in NGC Application for Neural Tissue Engineering

NGCs have undergone tremendous development throughout the last two decades and are now offering the possibility of providing an alternative to the current treatment options. With our increased understanding of the biology of nerve injury and repair, new designs that meet the demands of molecular and cellular processes are being generated. Typically, these aim to create a bridge across the lesion while bringing necessary support and supplying appropriate cues for tissue regeneration [86]. However, much uncertainty still exists about whether NGCs provide superior outcomes over traditional therapeutic modalities.

### 5.1. Personalizing and Automating the Fabrication Process

Medical imaging of the injury site can allow researchers to determine the size and characteristics of the injury site and then use that medical imaging data to create a patient-specific NGC to suit the injury. At present, special attention is given to developing an automated process such as rapid prototyping that incorporates patient-specific features as a crucial aspect for the fabrication of NGC [91,92]. In this regard, there are two main limitations: lack of instruction for creating tissue blueprints and inadequate streamlined production processes. Fabrication of NGCs that can match the patient-specific requirements in injured nerves, local vasculature, and fascicular architecture could become a major pathway in transplantation strategies in the near future [8]. Machine intelligence is a promising concept that can considerably advance current NGC manufacturing processes [8,93]. In this regard, studies are ongoing to demonstrate the great potential of deep learning technology in bio fabrication, such as selecting patient-specific NGC characteristics based on parenchymal composition, nerve types, modes of injury, and genetic variations. For a more detailed review of these concepts, see [8]. Nonetheless, the application of machine learning in optimizing NGCs is at an early stage of development and requires further extensive studies.

### 5.2. Enhancing Cell Survival and Integration

While various cell types such as stem and progenitor cells, or glial cells have been extensively considered in preclinical trials, the therapeutic approaches yielded limited success due to low integration, survival rates, synaptogenesis, and function of the transplanted cells upon implantation. To overcome the poor survival of cells within NGCs after transplantation, strategies are needed that help create a niche for the cells that promote cell interaction, adhesion, and survival. In this regard, large fibre-based constructs, such as nanofibrous scaffolds incorporated with trophic factors and/or pharmacological agents, have been shown to be highly efficient in cell transplantation via mimicking the endogenous nervous tissue microenvironments. As such, a synergistic effect of combining cell delivery, drug or gene delivery, and material design, as well as the optimum balance of delivered agents without provoking immune responses, may achieve satisfactory functional recovery in the damaged area [44,86]. However, this requires considerable testing of the numerous parameters to determine how the complex interactions can benefit not only the transplanted cells but also endogenous cells.

## 6. Summary

The fibre-based NGCs provide new hopes to treat damaged nerves, especially over long gaps, by providing a nano-engineered environment and offering biomimetic structures similar to natural ECM. The desired cell response can be obtained by modifying the design parameters of electrospun fibres such as fibre diameter, alignment, density, biocompatibility, surface nanotopography, and surface chemistry [94]. These conduits also provide a vector for delivery cells and biochemical factors to generate a nourishing environment that leads to enhanced functional recovery. NGC material, morphological structure, topographical features, an adhesive surface are crucial aspects for NGC design and fabrication. Desirable properties of NGC can also be achieved by providing them with the appropriate combination of cells and neurotrophic factors, as well as biochemical and physical cues. However, there is still a significant amount of work to be performed in the different aspects of cellular machinery, tissue engineering, and surgical procedures to establish NGC’s superiority over conventional autografts for treating large-gap neural injuries. The highest potential lies in the combination of tissue engineering and machine intelligence approaches to optimize the performance of NGC.

The dream of re-establishing the damaged nerve with complete functionality in patients with large-gap injuries can be fulfilled by utilizing a collaborative effort in which medicine, neuroscience, nanotechnology, biology, computer science, and engineering have been carefully combined. This combination can offer a comprehensive biomanufacturing process, leading to well established multidisciplinary platforms with optimal physicochemical characteristics which can concisely mimic the natural ECM and meet the requirements in the target area.

## Figures and Tables

**Figure 1 pharmaceutics-14-00219-f001:**
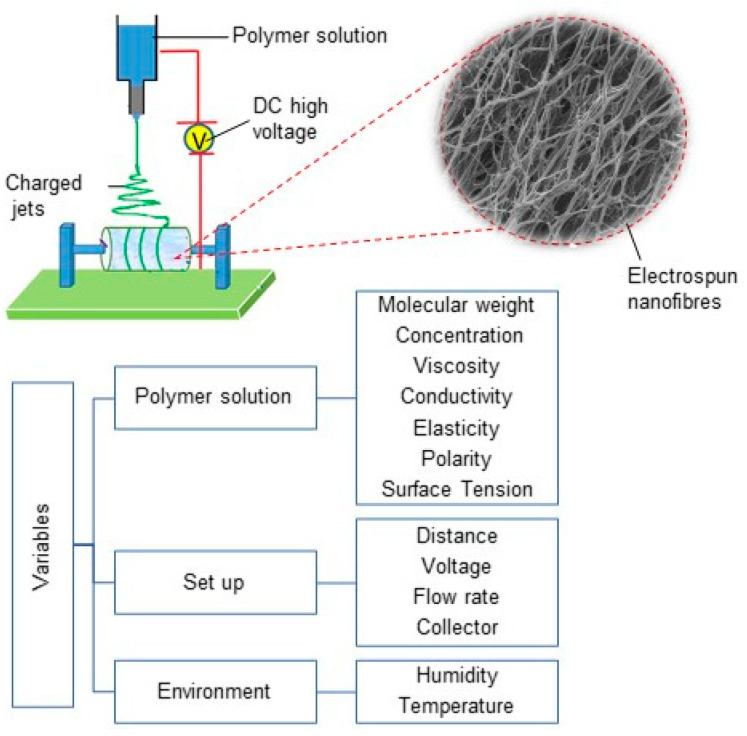
Electrospinning set-up diagram, electrospun nanofibres and different variables in electrospinning process.

**Figure 2 pharmaceutics-14-00219-f002:**
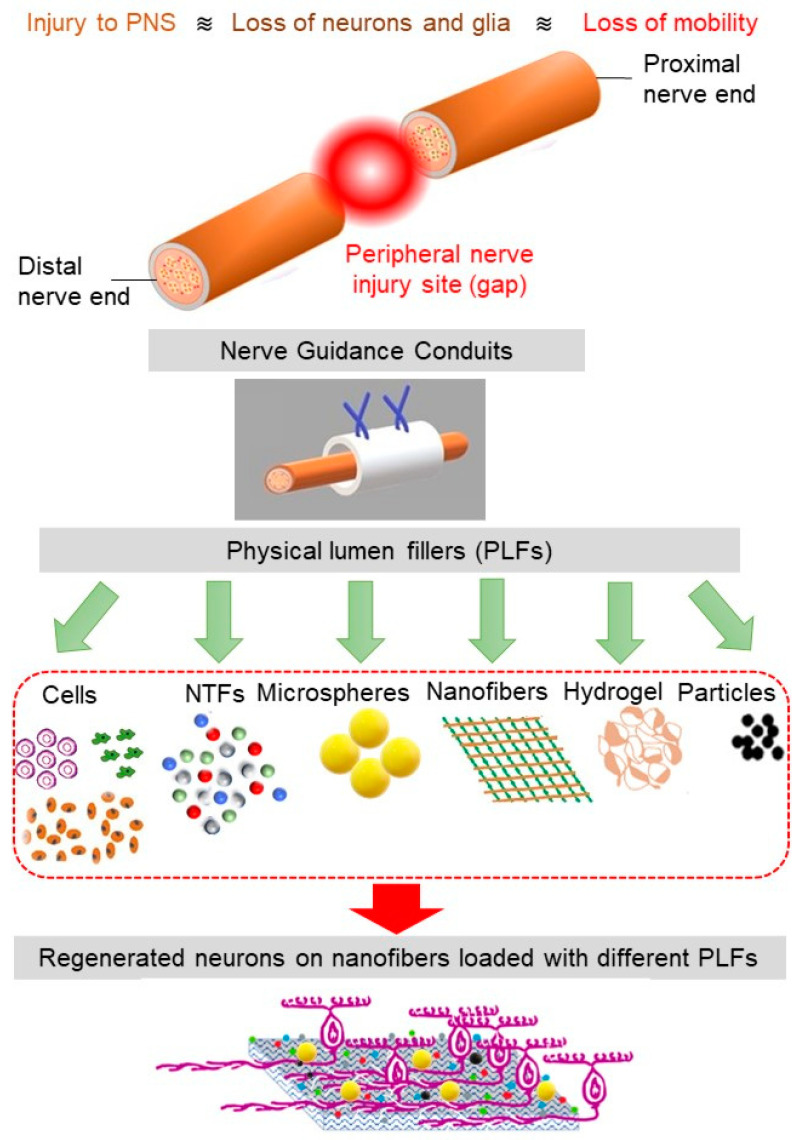
The schematics of PNS damage, NGC, and various physical lumen fillers (PLFs). The nanofibers loaded with the appropriate combination of PLFs facilitate cell regeneration.

**Figure 3 pharmaceutics-14-00219-f003:**
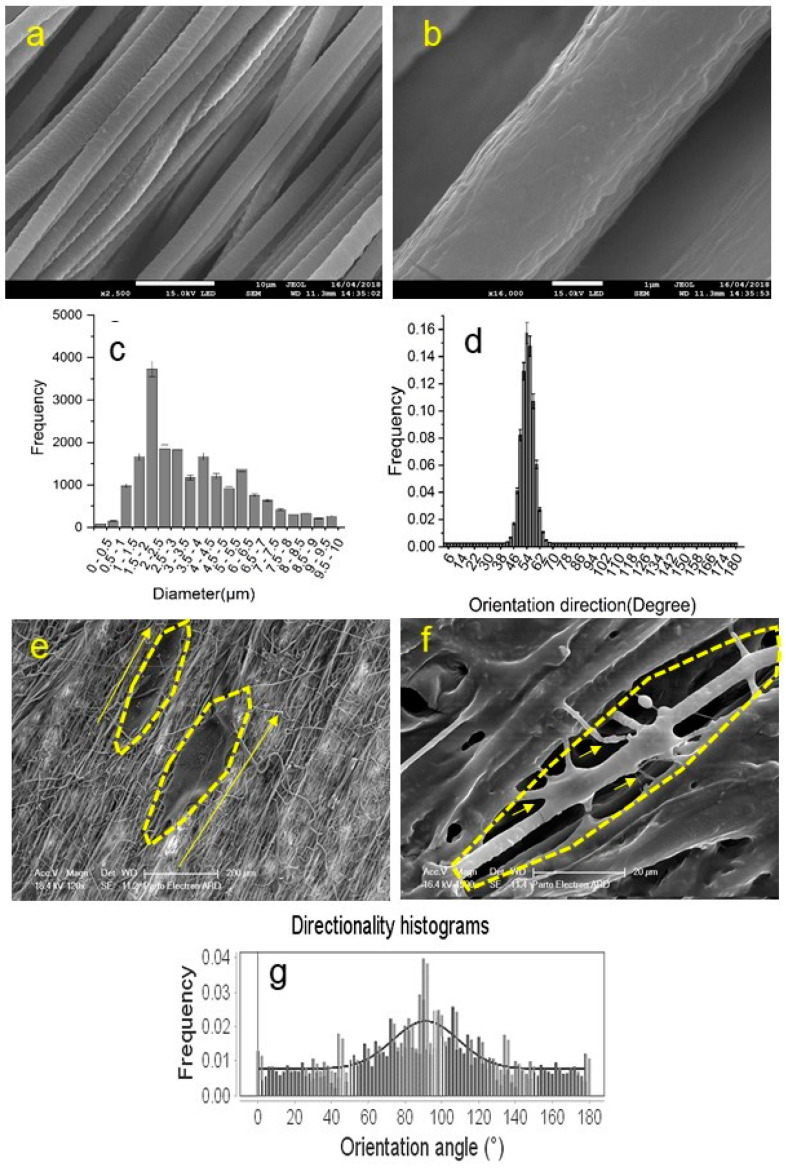
(**a**) SEM images of aligned fibrous scaffolds (scale bar: 10 µm); (**b**) SEM images of aligned fibrous scaffolds (scale bar: 1 µm); (**c**) the quantitative distribution of fibre diameters; (**d**) The orientation direction distribution of the scaffold fibres; (**e**) the SEM image of the neurons on the aligned scaffold (scale bar: 200 µm); (**f**) the SEM image of the neurons on the aligned scaffold (scale bar: 20 µm); and (**g**) the directional histogram of cells on the aligned scaffolds obtained by ImageJ^®^.

## Data Availability

Data sharing not applicable.

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
