# Peer review of "Advances in Electrospun Nerve Guidance Conduits for Engineering Neural Regeneration"

_pharmaceutics, 2022, doi:10.3390/pharmaceutics14020219_

Round 1
Reviewer 1 Report
I reviewed the article entitled „Advances in electrospun nerve guidance conduits for engineering neural regeneration“. The content of the article is well structured. However, there are missing important information listed below.
- Specific required properties of NGCs structure (subchapter 1.3, 1.7) should be mentioned (especially size of the fabricated NGCs which makes the fabrication quite challenging).
- In the subchpater 1.3, there is mentioned porosity that is difficult to measure in this compex structure which should be discussed as well.
- In the subchpater 1.7, details of fabrication of tubular graft as well lumen fillers should be described. Also in other subchpaters, extension of electrospinning procedures for fabrication of NGCs would be helpful to describe in more detail since the article is devoted to electrospun materials used as NGCs.
- Table 1 should be extended for other studies mentioned in the text. I recommend to omit ES parameters since there are many other parameters that could be found in references.
- The authors should also discuss repeatability of electrospinning process for fabrication of NGCs.
Based on the above mentioned points, I recommend revision of the text for further consideration of publishing.
Reviewer 2 Report
This is a timely and interesting review, which is quite well written and has no major issues to be addressed. The first sentence of the introduction reads “Millions of people worldwide are affected by NEURODEGENERATIVE diseases damaging the peripheral nervous system (PNS).” And the second sentence reads “PNS DAMAGE can happen during traffic accidents and other trauma, resections of tumours and/or adverse iatrogenic effects of the surgery.” Both sentences are right, but author should clearly differentiate between degeneration vs damage/injury. As far the PNS/peripheral nerve is concerned, the leading cause of neuropathy is probably injury.
Besides surgical procedures, recent studies highlight the genetic and adoptive cell transfer strategies as promising means for accelerating nerve regeneration. Including these advancements will enrich this manuscript.
Reviewer 3 Report
This work presents a review about the nerve grafting and replacement through electrospun nanofibers. The review is intensive in the part of material and technique, but it is still lacking of other important features as follows:
1- There is no information about diameters of electrospun nanofibers. This is crucial in nerve connection.
2- The manuscript must have some minor photos about nanofibers, nerve grafting, connection/replacement process. Copyrights clearance has to be obtained.
3- Technique of nerve signal flow after connecting the fibers still needs more explanation. How do nerve signals flow normally through connected fibers?
4- Piezo/capacitive technique of signal flow is not presented in the review along with its materials such as spider silk or PVDF.
5- One serious drawback in the manuscript is the lack of analytical results. The authors have to mention different statistical/analytical results about the improvement of nerve system after injury and after treatment with fibers.
6- Some parts still need detailed explanations such as magnetic properties and its impact in nerve cells grafting/regrowth.
Round 2
Reviewer 3 Report
All my comments have been well addressed.